# The β-Blocker Carvedilol Prevented Ultraviolet-Mediated Damage of Murine Epidermal Cells and 3D Human Reconstructed Skin

**DOI:** 10.3390/ijms21030798

**Published:** 2020-01-25

**Authors:** Mengbing Chen, Sherry Liang, Ayaz Shahid, Bradley T. Andresen, Ying Huang

**Affiliations:** Department of Pharmaceutical Sciences, College of Pharmacy, Western University of Health Sciences, Pomona, CA 91766, USA

**Keywords:** carcinogenesis, chemoprevention, β-blocker, carvedilol, ultraviolet, H_2_O_2_, ROS, CPD, PGE_2_

## Abstract

The β-blocker carvedilol prevents ultraviolet (UV)-induced skin cancer, but the mechanism is unknown. Since carvedilol possesses antioxidant activity, this study investigated whether carvedilol prevents oxidative photodamage of skin, a precursor event in skin carcinogenesis. The effects of carvedilol, metoprolol (a β-blocker without antioxidant property), and 4-hydroxycarbazole (4-OHC, a carvedilol synthesis intermediate and a free radical scavenger) were compared on UV- or H_2_O_2_-induced cell death and reactive oxygen species (ROS) production in murine epidermal JB6 P+ cells. Although carvedilol attenuated cell death, metoprolol and 4-OHC failed to show protective effects. As expected, increased cellular ROS induced by H_2_O_2_ or UV was abolished by carvedilol and 4-OHC, but not by metoprolol. Consistently, carvedilol attenuated the formation of UV-induced cyclobutane pyrimidine dimers (CPDs) and release of prostaglandin E_2_ in JB6 P+ cells. Carvedilol’s activity was further confirmed in full thickness 3D human reconstituted skin, where carvedilol attenuated UV-mediated epidermal thickening, the number of Ki-67 and p53 positive cells as well as CPD formation. Based on pathway-specific Polymerase Chain Reaction (PCR) Array analysis, carvedilol treatment in many cases normalized UV-induced expression changes in DNA repair genes. Thus, carvedilol’s photoprotective activity is not attributed to β-blockade or direct ROS-scavenging capacity, but likely via DNA repair regulation.

## 1. Introduction

The β-adrenergic receptor (β-AR) antagonists, commonly called β-blockers, are traditionally used for cardiovascular disorders [1]. These agents act by binding to the β-ARs and thereby inhibiting catecholamines from triggering the body’s “fight or flight” response to stress [2]. Carvedilol is a receptor subtype non-selective β-blocker with α-AR blocking and antioxidant properties [3,4,5]. Recent data demonstrate that carvedilol prevents malignant transformation and carcinogenesis induced by epidermal growth factor (EGF), chemical carcinogens, and UV [6,7,8,9]. However, the cancer preventive mechanism for carvedilol remains unknown. Notably, not all β-blockers had effects on EGF-induced transformation [6,9]. Since all β-blockers inhibit β-ARs, the stand-out cancer preventive activity of carvedilol suggests that the mechanism might be associated with a unique property of carvedilol and that the β-ARs may not be the direct target for carvedilol’s cancer preventive activity [10]. 

Carvedilol is a unique β-blocker with antioxidant and pleiotropic activity [11,12]. It exhibits two distinct antioxidant properties [11]: first, it is a chemical antioxidant with an ability to bind to and scavenge reactive oxygen species (ROS); second, it has pharmacological activity that suppresses ROS production. Since oxidative stress plays an essential role in UV-induced skin carcinogenesis and antioxidants have been proposed as skin cancer preventive agents [13,14,15], we hypothesized that the antioxidant property of carvedilol is responsible for its skin cancer preventive effects. Supporting this hypothesis, carvedilol has been reported to have an inhibitory activity against hydroxyl radical production [3], lipid peroxidation [16], and apoptosis [17], but has not been evaluated in UV-associated skin models. It is also unknown whether carvedilol’s ROS-scavenging, ROS-suppressive activity, or both are involved in its cancer preventive activities. 

The present study first examined whether carvedilol could protect murine epidermal cells against UV or hydrogen peroxide (H_2_O_2_)-induced oxidative damage, a known precursor event leading to skin carcinogenesis [18]. Carvedilol’s protective activity was compared with metoprolol, a β-blocker without proven antioxidant property [19], and an intermediate compound for carvedilol synthesis, 4-hydroxycarbazole (4-OHC), which possesses a chemical antioxidant activity [20]. In previous studies, metoprolol and 4-OHC did not show cancer preventive efficacy when compared with carvedilol in vitro and in vivo [7,9]. Furthermore, to confirm carvedilol’s photoprotective activity, a reconstituted 3-dimensional (3D) human skin model was used. In the 3D model, carvedilol was effective in modulating UV-induced photodamage biomarkers and gene expression profiles associated with DNA damage and repair response signaling. Our results indicate that the skin cancer preventive effects of carvedilol may be partially mediated by its ROS-suppressing activity, but not via direct ROS-scavenging activity. A novel activity of carvedilol in regulating DNA repair was also indicated by our data. 

## 2. Results

### 2.1. Protective Effects of Carvedilol on ultraviolet (UV)-Induced Epidermal Cell Death

Exposures of JB6 P+ cells to 25–400 mJ/cm^2^ UV resulted in a dose-dependent decrease in cell viability. As 25 mJ/cm^2^ UV was the minimum dose inducing significant cell viability changes, it was therefore chosen for further study. Carvedilol at 0.1 to 5.0 μM dose-dependently prevented UV-induced cell death with 5.0 µM displaying no significant difference than the controls (Figure 1A). Metoprolol failed to show any protective effect (Figure 1B). 4-hydroxycarbazol (4-OHC) also failed to show any protective effects (Figure 1C). Carvedilol (10 μM), but not metoprolol, inhibited UV-induced ROS formation in a similar manner to the positive control resveratrol (50 μM), whereas 4-OHC had an intermediate effect (Figure 1D). The UV-associated viability and ROS studies confirm previous studies that carvedilol may not act through the β-Ars in skin cells [10], but cannot exclude the possibility that ROS suppression may be its protective mechanism against UV-induced epidermal cell death.

### 2.2. Protective Effects of Carvedilol on H_2_O_2_-Induced Epidermal Cell Death

Exposure of the skin to UV is known to induce oxidative stress via the over-production of H_2_O_2_ [21]. Therefore, we next determined whether carvedilol prevents the oxidative damage induced by H_2_O_2_. H_2_O_2_ dose-dependently decreased the cell viability of the JB6 P+ cells. When the cells were pre-treated with carvedilol (0.1–5.0 µM) for 2 h before the addition of 100 μM of H_2_O_2_ and then co-treated with carvedilol and H_2_O_2_ for an additional 24 h, there was an increase in cell viability in a dose dependent manner (Figure 2A). These effects were not observed when the cells were treated with the same concentrations of metoprolol (Figure 2B) or 4-OHC (Figure 2C). However, 4-OHC, although not statistically different, was more efficacious in reducing H_2_O_2_-induced ROS than carvedilol and resveratrol (Figure 2D). Therefore, the protective effect of carvedilol cannot be attributed to the direct scavenging of ROS as 4-OHC would have similar effects on cell viability if the chemical antioxidant was the mechanism involved in the carvedilol’s protective effect.

### 2.3. Effects of Carvedilol on UV-Induced Epidermal Cyclobutane Pyrimidine Dimer (CPD) Formation and PGE_2_ Secretion

Exposure of JB6 P+ culture to 25 mJ/cm^2^ UV radiation resulted in the time-dependent formation of CPD (Figure 3A). Treatment with carvedilol for 24 h resulted in a partial, but non-statistically significant, reduction in CPD levels (Figure 3A). Exposure of JB6 P+ cells to 25 mJ/cm^2^ UV statistically increased PGE_2_ secretion into the culture media by 1.8 ± 0.5 fold (Figure 3B). Treatment with 5 µM carvedilol for 24-h post UV exposure statistically reduced UV-induced PGE_2_ secretion into the culture media to a level that was not statistically differentiable from the control; the fold difference from the control was 1.12 ± 0.37. However, carvedilol also statistically reduced the basal level of PGE_2_ secretion by 50%; the fold difference from the control was 0.49 ± 0.15. The fold increase from the carvedilol-treated basal PGE_2_ level (No UV, CAR) to the UV-induced level (UV, CAR) was 2.3 ± 0.4 fold, which is similar to the UV-induction in the control. The CPD and PGE_2_ data suggest that the mechanism is not simply via scavenging ROS via cell signaling regulation. Furthermore, the in vitro cell culture CPD data mirrored, but was not as robust as studies with mouse skin in vivo [7]. 

### 2.4. Effects of Carvedilol on UV-Induced Damage of 3D Reconstructed Human Skin

To confirm the photoprotective effect of carvedilol, a reconstituted human full thickness skin model, EpiDerm^TM^ FT-400, was used. This model contains both epidermal and dermal layers in 3D and has been reported to exhibit a similar response to UV as human skin [22]. Histological analysis with hematoxylin and eosin (H&E) staining was performed to determine the morphological changes induced by UV (60 mJ/cm^2^) and carvedilol (5 µM) (Figure 4A). The results showed a modest trend of increased epidermal thickness in the skin samples treated with UV (Figure 4A, top row): UV treatment increased epidermal thickness by 12%, and carvedilol treatment completely prevented UV-induced thickening (average of ten randomly selected epidermal thickness measurements; *P* < 0.05; *n* = 1). Analogous to the H&E data, immunohistochemical (IHC) analysis indicated a 50% increase in Ki-67 positive cells within the basal membrane of the epidermis of the cultures treated with UV (Figure 4A, middle row); carvedilol attenuated UV-induced Ki-67 staining to a 30% increase over the control (*P* > 0.05; *n* = 1). Consistently, UV exposure increased the number of p53 positive cells in the epidermis by 30% (Figure 4A, bottom row) while carvedilol treatment attenuated UV-induced p53 staining to a 13% increase over control (P>0.05; *n* = 1) (average of three randomly selected fields counted for Ki-67 or p53 positive cells). Although UV radiation statistically increased the CPD production in the 3D human skin culture, 5 μM carvedilol did not reduce CPD (*n* = 2) (data not shown). Unlike the cell culture of JB6 P+, UV failed to statistically increase PGE_2_ production in the 3D culture, and 5 μM carvedilol had no effect on PGE_2_ production (*n* = 3) (data not shown). 

### 2.5. Effects of UV and Carvedilol on Expression of DNA Damage Signaling Genes by RT2 Profile Polymerase Chain Reaction (PCR) Array

Since the effects of UV and carvedilol on the 3D skin culture were modest, to identify the molecular events of carvedilol on UV-mediated regulation of gene expression, a Human DNA Damage Signaling RT^2^ Profile PCR array with 84 genes was utilized. The 60 mJ/cm^2^ UV irradiation downregulated 35 genes by at least 20%. Carvedilol treatment attenuated the UV-induced decrease of all 35 genes to different degrees, with 20 genes normalized to the control levels (Figure 4B, left panel). Interestingly, many of these normalized genes are involved in the DNA repair response to UV or other types of radiation (e.g., *RAD51B*, *XRCC1*, *RPA1*, *MSH3*, *ATRX*, *OGG2*, etc.). UV irradiation upregulated 11 genes by at least 20%, and carvedilol attenuated all 11 genes with six genes reduced by 30% or more (Figure 4B, right panel). The PCR array data suggest that the photoprotective effects of carvedilol may be partly mediated by altered DNA repair.

### 2.6. Effects of Higher Doses of Carvedilol on UV-Induced Damage in Reconstructed Human Skin

Since 5 μM carvedilol did not result in a protective effect on UV-induced CPD formation in the 3D culture, we repeated the experiment using a separate batch of reconstructed human skin culture irradiated with 200 mJ/cm^2^ UV (for stronger effects of UV) and treated with higher doses (20 or 100 μM) of carvedilol. Additionally, to avoid any damaging effect of acetone, carvedilol was dissolved in the culture media. An increased formation of CPD was detected in the UV-irradiated skin samples while carvedilol dose-dependently attenuated UV-induced CPD formation, although this was not statistically significant (*n* = 3) (Figure 5A) 

Since the repair enzymes for the DNA damage marker 8-hydroxy-2'-deoxyguanosine (8-OHdG), OGG1, and MUTYH [23] were altered by UV/carvedilol (Figure 4B), 8-OHdG level was evaluated in the 3D skin culture. As can be seen in Figure 5B, the 8-OHdG levels were variable in the control and UV-irradiated samples; paradoxically, UV did not increase the 8-OHdG levels over the control samples on average. Likewise, 20 µM carvedilol had no effect, while 100 μM carvedilol essentially ablated the levels of 8-OHdG. 

To examine a third line of cellular responses to UV irradiation, apoptosis was examined via Poly (ADP-ribose) polymerase (PARP) western blot. From the cleaved PARP level, although with a high variation, 20 μM carvedilol showed a trend of reducing UV-induced apoptosis in the 3D culture (Figure 5C) (*p* > 0.05). However, 100 μM carvedilol did not show any protective effects. 

To verify the effects of UV and carvedilol on DNA repair genes, select genes involved in DNA repair were analyzed by Reverse Transcriptase PCR (RT-PCR). Contrary to lower dose UV irradiation (Figure 4B), 200 mJ/cm^2^ UV treatment failed to decrease the mRNA expression of the DNA repair enzymes (*RPA1*, *GADD45A*, *XRCC1*, and *OGG1*), but showed a trend of expression increase (*p* > 0.05), while carvedilol reduced the expression, although not significantly (Figure 6A–D). 

Proinflammatory biomarkers (IL-6, IL-1β, TNF-α, and COX-2) known to be involved in UV-induced skin inflammation were also examined. Although these markers are known to change after UV irradiation, the conditions herein failed to demonstrate any significant UV effects, thus, preventing conclusions from being drawn regarding the carvedilol-mediated effects (Figure 6E for IL-1β; data not shown for other genes). Although 20 μM carvedilol reduced IL-1β expression, 100 μM greatly increased it. Furthermore, 200 mJ/cm^2^ UV-irradiation also failed to increase the PGE_2_ levels in culture media, while 100 μM carvedilol greatly increased the secretion of PGE_2_ (by 66 folds) (Figure 6F). 

## 3. Discussion

Consistent with carvedilol’s known antioxidant activity, it prevented epidermal cell death induced by UV or H_2_O_2_ (Figure 1 and Figure 2). Two potential mechanisms for carvedilol’s protective effects were examined in these experiments: β-AR blockade and ROS scavenging activity. Metoprolol was used as a control β-blocker without anti-transformation activity [9] and without proven antioxidant property [19], but is a commonly used β-blocker. 4-OHC is an intermediate compound in carvedilol synthesis that acted as a free radical scavenger (i.e., chemical antioxidant) [20]. Carvedilol-mediated protective effects were unique as neither UV- nor H_2_O_2_-mediated reductions in cell viability were altered by metoprolol or 4-OHC. This result is consistent with previous reports that only carvedilol, but not 4-OHC and metoprolol, inhibited UV- or EGF-induced NF-κB and AP-1 promoter activity and JB6 P+ neoplastic transformation [7,9]. Resveratrol, used as a positive control in the ROS assay, is a potent antioxidant with relatively poor direct scavenging activity [24]. Resveratrol was not included in the viability assay due to previously reported effects against UV-mediated skin damage and carcinogenesis [25]. Thus, these data strongly suggest that the protective effects of carvedilol, similar to resveratrol, are not mediated by direct ROS scavenging, but by a capability associated with the suppression of ROS generation. 

Two pathways known to be involved in skin carcinogenesis are UV-induced CPD formation (a DNA damage marker) and PGE_2_ production (an inflammatory marker). Previously, it has been reported that carvedilol reduced DNA damage [7,8] and inflammation [11]. However, the data in the JB6 P+ cells and the 3D skin culture (Figure 3A and Figure 5A) were not as robust as the data obtained in mice [7]. The lesser effect in isolated cell culture for carvedilol’s effect on CPD suggests that there is a tissue microenvironment-dependent effect that is partially lost in vitro. Supporting this hypothesis is the PGE_2_ production. Previous studies indicate that carvedilol reduced PGE_2_ levels and overall inflammatory reactions in mice [26]. Topical carvedilol treatment attenuated UV induced pro-inflammatory gene expression such as Cox-2 in mouse skin [7], whereas the PGE_2_ changes in JB6 P+ cells and the 3D skin culture treated with UV/carvedilol (Figure 3B and Figure 6F) were modest or failed to work. Although not as robust as in vivo, carvedilol’s cancer preventive activity can be partially mediated by reducing DNA damage and inflammation. Since DNA damage is considered as a consequence of ROS production [27] and inflammation can also be driven by ROS [28], the antioxidant properties of carvedilol play a role in mediating these effects. 

To overcome the limitation of monoculture, a commercially available 3D human skin culture model was utilized. Unfortunately, for both experiments conducted in this model, the sample sizes were low (*n* = 2~3). The results indicate that the UV (60 mJ/cm^2^) or carvedilol (5 μM) slightly altered skin thickness, Ki-67, and p53 expression. The UV dose (60 mJ/cm^2^) was selected based on previous reports using the same 3D model [29,30]. Due to the modest effect, the UV dose was increased in the second experiment to 200 mJ/cm^2^. However, there was still no significant increase in PGE_2_ production or inflammatory gene expression when the culture was treated with a higher UV dose (Figure 6E,F). One explanation is that the UV doses were not sufficiently high, since a previous study demonstrated PGE_2_ secretion with 300 mJ/cm^2^ of UVB [31]. This could also be the reason for the lack of a statistically significant effect on the expression of proinflammatory genes (e.g. IL-1β). Although UV did not robustly increase the level of the proinflammatory genes, treatment with carvedilol at 20 μM attenuated the induction of these genes. However, 100 μM carvedilol promotes inflammation, as it greatly enhanced the proinflammatory genes and PGE_2_ level (Figure 6E,F). Based on the 3D data, it appears that a higher dose of carvedilol is required to attenuate DNA damage; however, if the doses are too high (e.g., 100 μM), carvedilol promotes inflammation. Thus, the optimal therapeutic doses of carvedilol as a anticancer agent remain to be determined. 

A surprising finding derived from the PCR array assay was that at a lower dose (60 mJ/cm^2^), UV downregulated DNA repair genes (Figure 4), while a high dose (200 mJ/cm^2^) showed the opposite effects (Figure 6), although in both experiments carvedilol normalized the UV effect. Previous studies have demonstrated that UV or other carcinogens decreased expression or activity of DNA repair enzymes: tobacco smoke reduced DNA repair enzymes, XPC and OGG1, in lung tissue [32,33]; infrared radiation reduced GADD45a in cultured human fibroblasts [34]. In other studies, UV increased expression of DNA repair enzymes GADD45a [35], XPA, XPC, and RPA1 [36]. Thus, the effects on DNA repair is dependent on the UV dose and time point. Although we cannot draw a complete conclusion from these gene expression-based studies, carvedilol may be involved in DNA repair pathways to attenuate UV-mediated DNA damage. Further studies are needed to determine whether such an effect is unique for carvedilol, under a wider range of UV and drug doses. A transcriptome analysis is needed to decipher global gene expression changes, due to the limitation of PCR array including only 84 genes. 

Despite the limitations, the collected results support the notion that the skin cancer chemopreventive effects of carvedilol are primarily due to a mechanism that is independent of β-adrenergic blockade and direct ROS scavenging activity. Regulating DNA repair may represent a novel mechanism responsible for carvedilol’s cancer preventive activity. 

## 4. Materials and Methods 

### 4.1. Compounds

Carvedilol and metoprolol were purchased from Tocris Bioscience (Minneapolis, MN). 4-OHC and H_2_O_2_ were from Sigma-Aldrich (Saint Louis, MO, USA).

### 4.2. Cell Culture

The mouse epidermal cell line JB6 Cl 41-5a (JB6 P+), a tumor promoter sensitive subline, was purchased from American Type Culture Collection (ATCC, Manassas, VA). The cells were cultured in Eagle’s minimum essential medium (EMEM) (ATCC) containing 4% (*v*/*v*) heat-inactivated fetal bovine serum (FBS) and 1% penicillin/streptomycin, in a humidified atmosphere of 37 °C and 5% CO_2_/95% air. 

### 4.3. UV Light Source

UV lamps emitting a mixture of UVB, UVA, UVC, and visible light (UVP, Upland, CA, USA) were used to irradiate the 2D and 3D cultures. Stable power output (mW/cm^2^) was measured using a UVX radiometer (UVP) coupled with a sensor with a calibration point of 310 nm (UVP), and exposure time was calculated using the following formula: dose (mJ/cm^2^) = exposure time (s) × output intensity (mW/cm^2^). 

### 4.4. Trypan Blue Exclusion Assay

Cells were seeded in 12 well plates that were irradiated with UV in PBS and after UV, the PBS was immediately replaced with media containing the test drugs. For H_2_O_2_ treatment, the cells were pretreated with drugs for 2 h before adding H_2_O_2._ The plate was further incubated with the drugs for 24 h before the cells were collected and mixed with a 1:5 dilution of Trypan blue (Gibco, Gaithersburg, MD, USA). The number of viable cells was determined by counting the non-stained cells using dual chamber cell counting slides and an automatic cell counter (BioRad, Hercules, CA, USA).

### 4.5. Cellular Reactive Oxygen Species (ROS) Assay

Intracellular ROS was determined using the H2-DCF-DA dye/probe (Invitrogen, Carlsbad, CA, USA). Cells were seeded in a 12 well plate and testing drugs were added 2 h before the cells were treated with PBS containing 10 µM H2-DCF-DA, incubated for 30 min before UV exposure, or in full culture media containing H_2_O_2_, incubated for 30 min. After incubation, the cells were trypsinized and the fluorescence was measured in the flow cytometer (Accuri C6, Accuri, Ann Arbor, MI, USA; Ex 488 nm; FL-1 530 nm +/− 15; 10,000 events).

### 4.6. Slot Blot Analysis for CPD and 8-Oxoguanine (8-OHdG)

Genomic DNA was isolated from the cells or the 3D skin by using QIAamp DNA Mini Kit (Qiagen, Redwood City, CA). The DNA samples (100 ng for CPD; 1 μg for 8-OHdG) were vacuum-transferred to a nitrocellulose membrane (0.45 Micron, Thermo Scientific) using a Bio-Dot SF microfiltration apparatus (Bio-Rad, Hercules, CA). CPDs were detected by immune-slot blot using anti-CPD monoclonal antibody (Kamiya, Seattle, WA) or 8-OHdG antibody (E-8) (Santa Cruz, Dallas, TX). Total DNA amounts loaded onto the membrane were visualized by SYBR-Gold (Invitrogen) staining and used to normalize the CPD or 8-OHdG values.

### 4.7. Enzyme-Linked Immunosorbent Assay (ELISA) Assay for PGE_2_

JB6 P+ cells were cultured in 12-well plates and pre-treated with carvedilol (5 μM) or the vehicle for 2 h and were then exposed to UV. After further incubation with the drug for 24 h post-UV, the culture media were collected from the plates and PGE_2_ levels determined by competitive ELISA, according to the manufacturer’s protocols (Cayman Chemical Co., Ann Arbor, MI, USA). 

### 4.8. 3D Human Reconstituted Skin Model

EpiDerm^TM^ FT-400 (EFT-400) in six well plates was purchased from MatTek Corporation (Ashland, MA). The skin culture was treated with UV and/drug, according to the protocols provided by the manufacturer. In brief, the culture was pretreated with carvedilol or the vehicle by applying 50 μL of carvedilol in acetone (5 μM) to the stratum corneum side and adding it the media (5 μM). The cultures were exposed to 60 mJ/cm^2^ of UVB in PBS. Immediately following radiation, PBS was replaced with media containing carvedilol, and topically treated with carvedilol and incubated for 24 h. A different batch of FT-400 was used to the repeat above experiments with increased UV dose and carvedilol concentrations: 100 μL of carvedilol in media (20 or 100 μM) was added to the stratum corneum side of culture as well as the culture media; the UV dose was 200 mJ/cm^2^.

### 4.9. RNA Isolation and Quantitative PCR (qPCR)

According to the protocols provided by the manufacturer, the skin cultures were homogenized and total RNA was isolated using Qiagen’s RNeasy Kit. cDNA was synthesized with the High Capacity cDNA Reverse Transcriptase Kit (Applied Biosystems, Grand Island, NY, USA). RNAs were examined for the expression of 84 genes involved in DNA damage signaling using RT^2^ Profile PCR Array Human DNA Damage Signaling Pathway (Catalog #: PAHS-029a, SABiosciences, Frederick, MD, USA). The PCR cycle condition was as follows: 94 °C for 4 min, followed by 40 cycles of 94 °C for 30 s, 62 °C for 30 s, 72 °C for 35 s, and then kept at 72 °C for 10 min. PCR was performed on a CFX96 real-time thermal cycler detection system (Bio-rad) and analyzed with the 2^−ΔΔCt^ with GAPDH as the reference gene. An artificial cut-off value of greater than a 20% change between the UV treated versus the untreated control was utilized to determine the UV effects. To examine the carvedilol effects, a cut-off value of greater than 30% change between the carvedilol plus UV group and the UV alone group was selected. Individual RT-qPCR analysis was conducted in the same method using gene specific primers for which sequences are available upon request. 

### 4.10. Histological and Immunohistochemical (IHC) Analysis

The 3D cultures were fixed in 10% neutral buffered formalin, paraffin embedded, sectioned using a microtome, and stained with H&E, according to commonly used procedures. The IHC stain was performed on a Ventana Discovery Ultra (Ventana Medical Systems, Roche Diagnostics, Indianapolis, IN, USA) automated IHC stainer. The slides were loaded onto the machine, and deparaffination, rehydration, endogenous peroxydase activity inhibition, and antigen retrieval were performed. Following primary antibody incubation, DISCOVERY anti-Rabbit HQ and DISCOVERY anti-HQ-HRP were incubated. The stains were then visualized with the DISCOVERY ChromoMap DAB Kit, counterstained with hematoxylin (Ventana). Primary antibody used: Ki-67: Clone# 30-9, rabbit monoclonal antibody; P53: Clone# DO-7, mouse monoclonal antibody (Ventana). Images were taken with a VENTANA iScan HT slide scanner and analyzed with a Ventana Image Viewer.

### 4.11. Western Blot Analysis

Protein was extracted from the 3D culture by homogenizing in Radioimmunoprecipitation (RIPA) lysis buffer (Santa Cruz, Dallas, TX, USA). A sample of 20 µg of protein was resolved by 10% sodium dodecyl sulfate- polyacrylamide gel electrophoresis (SDS-PAGE) and transferred to a nitrocellulose membrane. A solution of 1:1000 anti-PARP and anti-β-actin (Cell Signaling, Danvers, MA) antibodies in 5% non-fat milk was applied to the membrane and then exposed to goat anti-rabbit IgG-HRP (Cell Signaling Technologies) diluted 1:20,000 in 5% non-fat milk. Membranes were visualized with SuperSignal West Pico Chemiluminescent Substrate (Thermo Fisher). 

### 4.12. Statistical Analysis

Data in figures are expressed as mean ± SE or SD, plotted using GraphPad Prism version 7.0 (La Jolla, CA, USA). GraphPad Prism was used to analyze the data via a one-way or two-way analysis of variance (ANOVA) followed by Tukey-Kramer post hoc test. For all statistical analysis, the means were indicated to be statistically different when *p* < 0.05.

## 5. Conclusions

Carvedilol’s photoprotective activity is not attributed to its β-blockade or direct ROS-scavenging capacity, but probably to the DNA repair regulation.

## 6. Patents

This work has filed a provisional patent (U.S. Provisional Application No. 62/824,611). 

## Figures and Tables

**Figure 1 ijms-21-00798-f001:**
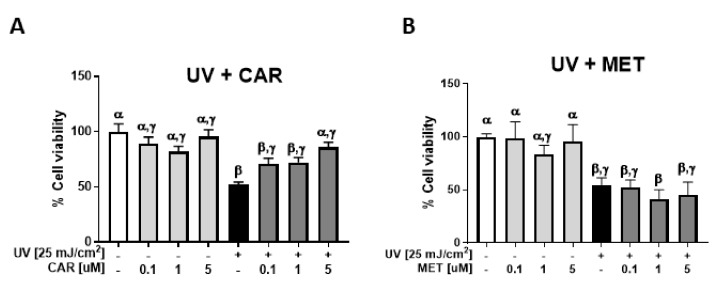
Effect of carvedilol, metoprolol, and 4-hydroxycarbazole (4-OHC) on ultraviolet (UV) radiation-induced reduction of cell viability and formation of reactive oxygen species (ROS) in JB6 P+ cells. JB6 P+ cells were treated with UV radiation at the dose of 25 mJ/cm^2^. After UV exposure, the cells were treated with the indicated concentrations of carvedilol (CAR) (**A**), metoprolol (MET) (**B**), or 4-OHC (**C**). After 24 h of incubation, the number of viable cells were counted using the Trypan blue exclusion assay. Data presented are the mean +/− SD of 3~6 independent experiments. The number of viable cells was normalized as the percentage of vehicle control. (**D**) JB6 P+ cells were treated with resveratrol (RES) (50 μM, positive control), carvedilol, metoprolol, or 4-OHC (10 μM each) for two hours and then loaded with PBS containing 10 µM H2-DCF-DA for 30 min. The cells were then exposed to UV irradiation (25 mJ/cm^2^) and incubated for 30 min. The DCF fluorescence levels were determined using flow cytometry. Each bar represents the mean +/− SE (*n* = 3). The bars with a different letter are significantly different from each other at the level of *P* < 0.05.

**Figure 2 ijms-21-00798-f002:**
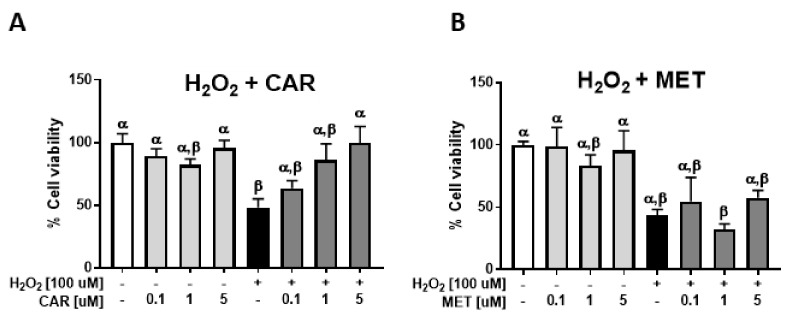
Effect of carvedilol, metoprolol, and 4-OHC on H_2_O_2_-induced reduction of cell viability and formation of ROS in JB6 P+ mouse epidermal cells. The cells were treated with H_2_O_2_ at the concentration of 100 μM. The cells were treated with indicated concentrations of CAR (**A**), MET (**B**) or 4-OHC (**C**) for two hours before H_2_O_2_ treatment and incubated with the drug and H_2_O_2_ for an additional 24 h before the quantification of viable cells using Trypan blue assay. Data presented are the mean +/− SD of 3~6 independent experiments. The number of viable cells was normalized as the percentage of vehicle control. (**D**) The cells were treated with resveratrol (RES) (50 μM, positive control), carvedilol, metoprolol, or 4-OHC (10 μM each) for two hours and then loaded with PBS containing 10 µM H2-DCF-DA for 30 min. The cells were then treated with 300 uM of H_2_O_2_ for 30 min. The DCF fluorescence levels were determined using flow cytometry. Each bar represents the mean +/− SE (*n* = 3). The bars with a different letter are significantly different from each other at the level of *P* < 0.05.

**Figure 3 ijms-21-00798-f003:**
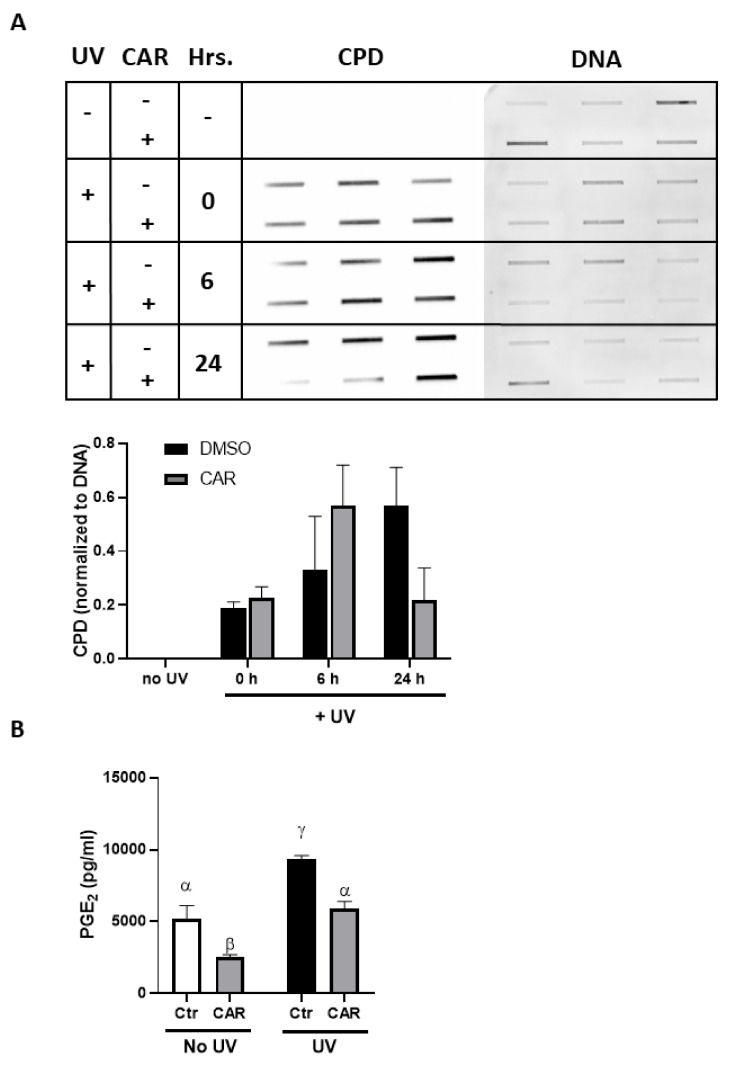
Effect of carvedilol on UV-induced CPD formation and PGE_2_ secretion in JB6 P+ cells. (**A**) The slot blot assay of CPD. Genomic DNA was isolated from the cultures irradiated with 25 mJ/cm^2^ UV and pretreated for 2 h and immediately post-UV treated with 5 μM carvedilol; the cells were harvested immediately (0 h), 6, or 24 h after UV exposure. CPD production was quantified and normalized by SYBR^TM^ Gold staining of the DNA. Each band represents a DNA sample from one cell culture from each treatment group. (**B**) PGE_2_ levels determined by enzyme-linked immunosorbent assay (ELISA) in culture media from untreated or UV-irradiated JB6 culture treated with dimethyl sulfoxide (DMSO, the vehicle) or carvedilol (5 μM). Data are presented as the means ± SE (*n* = 3). The bars with a different letter are significantly different from each other at the level of *P* < 0.05.

**Figure 4 ijms-21-00798-f004:**
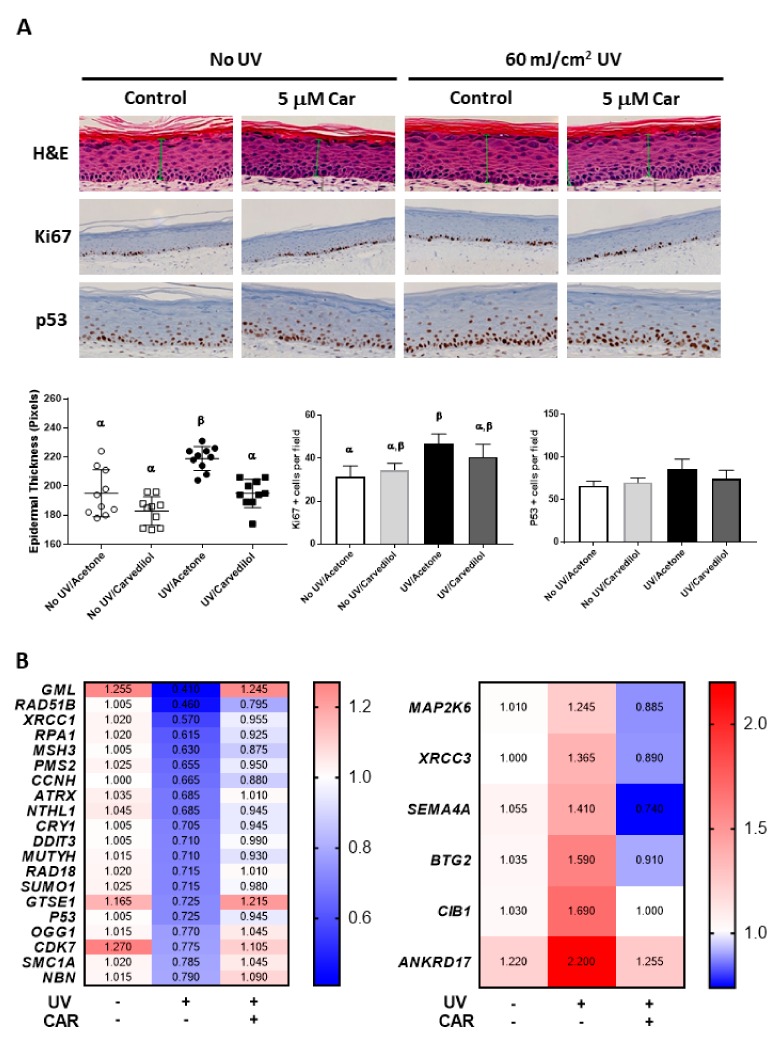
Effect of carvedilol on UV-mediated epidermal thickness, Ki-67, and p53 expression, and the expression of genes of DNA damage signaling in 3D human reconstituted skin culture (EpiDermTM FT-400). The reconstituted human full thickness skin was treated with UV (60 mJ/cm^2^), with or without carvedilol (CAR, 5 μM) for 24 h (see Materials and Methods for details). (**A**) H&E staining of EpiDermTM FT-400 3D human skin culture sections to quantify the epidermal thickness. The green arrows indicate the measurement of epidermal thickness. Skin sections were also analyzed with immunohistochemistry to detect Ki-67 or p53 positive cells. Representative sections of immunostaining are shown. Data were obtained from one sample from each treatment group (*n* = 1). The plots or bars with a different letter are significantly different from each other at the level of *p* < 0.05 as per two-way Analysis of variance (ANOVA) with Tukey’s post hoc test. (**B**) Profiling of DNA damage signaling genes by the RT^2^ Profile PCR array. RNA isolation and PCR array analysis were carried out on three treatment groups (vehicle control, UV control, and UV + carvedilol) (*n* = 2). The signal intensities of each protein were normalized by those of glyceraldehyde 3-phosphate dehydrogenase (*GAPDH*) and the fold change was calculated between the treated and vehicle control samples. A heat map was constructed using 2^−ΔΔ*C*t^ for the selected genes showing the highest fold changes.

**Figure 5 ijms-21-00798-f005:**
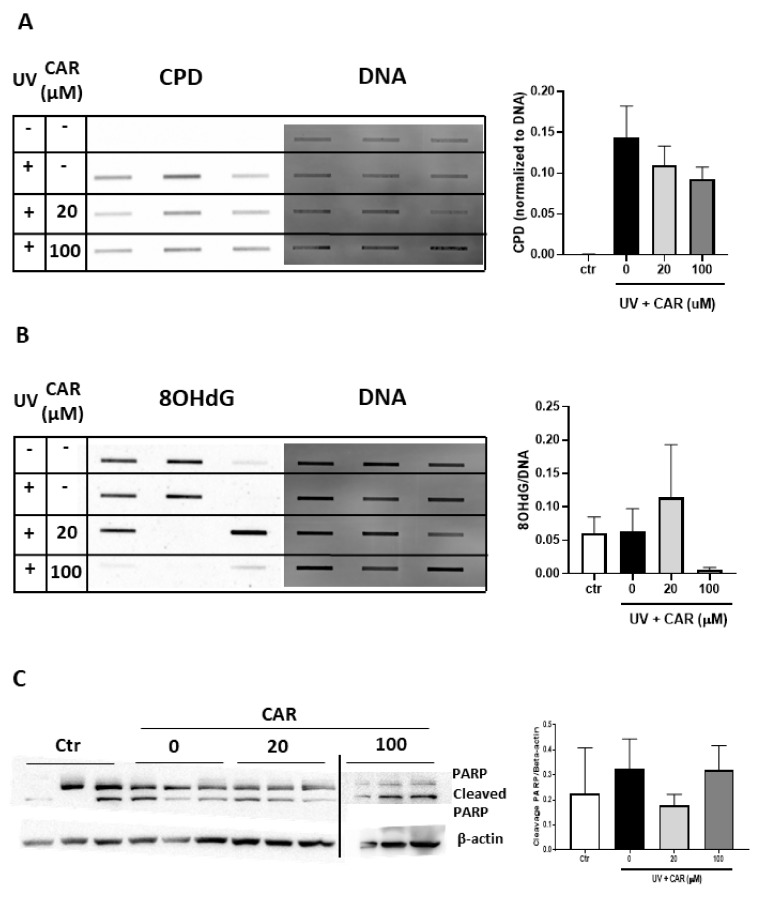
Effect of carvedilol on UV-mediated CPD and 8-OHdG formation and apoptosis in 3D human reconstituted skin (EpiDerm^TM^ FT-400). The reconstituted human full thickness skin was treated with UV (200 mJ/cm^2^), with or without carvedilol (CAR, 20, or 100 μM), for 24 h (see Materials and Methods for details). Genomic DNA was isolated from the cultures and analyzed with the slot blot assay with the CPD antibody (*n* = 3) (**A**) or the 8-OHdG antibody (*n* = 3) (**B**). (**C**) Western blot analysis of PARP, cleaved PARP, and β-actin. The blot images are cropped from two full-length blots and put together. The two gels were run and processed in parallel. Bar graphs represent the ratios of cleaved PARP/β-actin.

**Figure 6 ijms-21-00798-f006:**
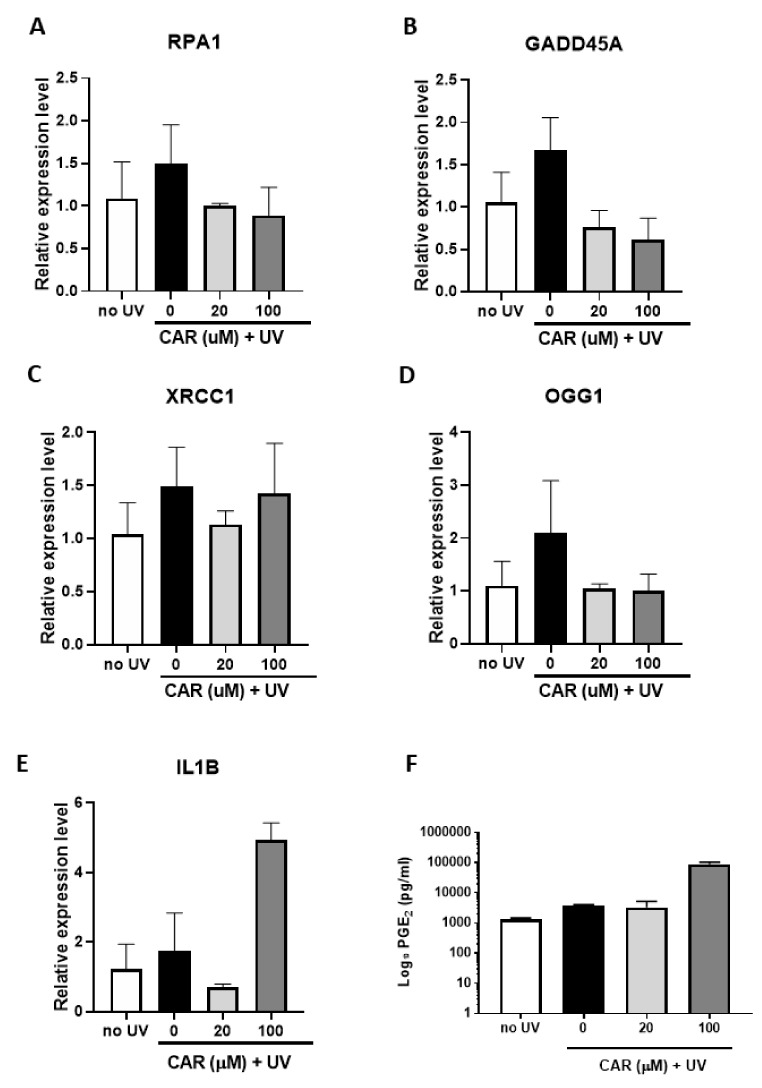
Effect of carvedilol and UV-mediated changes in expression of DNA damage/repair genes, proinflammatory marker *IL1B*, and PGE_2_ secretion in 3D human reconstituted skin (EpiDerm^TM^ FT-400). (**A**–**D**) qPCR results from 3D culture treated by UV 200 mJ/cm^2^ and/or carvedilol (20 or 100 uM in media) for *RPA1*, *GADD45A*, *XRCC1*, and *OGG1*. (**E**) qPCR results of *IL1B*. (**F**) PGE_2_ levels determined by enzyme-linked immunosorbent assay (ELISA) in culture media from untreated or UV-irradiated 3D culture treated with DMSO or carvedilol. GAPDH was used as a normalization control for qPCR and the data are expressed as mean +/− SE; *n* = 3.

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
