# Peer review of "The β-Blocker Carvedilol Prevented Ultraviolet-Mediated Damage of Murine Epidermal Cells and 3D Human Reconstructed Skin"

_ijms, 2020, doi:10.3390/ijms21030798_

Round 1

Reviewer 1 Report

The manuscript is original, interesting and well structured. Experiments are well designed and modern and relevant methods were used to study the protective potential effects of carvedilol on UV-radiation exposition. However, there are some mistakes and lack of information which indicates a lack of carefulness in the manuscript preparation. These errors and missing information should be addressed prior to publishing.

Major comments:

English quality should be improved throughout the manuscript – i recommend an expert revision of the language Introduction: Significant background information in the Introduction is missing. Improve metoprolol information and include 4-OHC. Sentence 15 to 18 (from Although carvedilol… to …by metropolol) should be rephrased to make sense. Sentence 23 should be rephrased so all verbs are in the same tense. Sentence between lines 223 and 225 should be rephrased to make sense. In lines 247 and 248 are missing references which are of critical importance given the sentence formulation. Probably the sentence should also be rephrased. Sentence in lines 258 and 259 is non-sense - rephrase Figures 1, 2 and 3 should be improved, place graphs in a 2x2 format Depth discussion is missing and key references should be necessarily included.

Minor comments:

Line 37: “s” from mechanism is missing Line 47: “its” instead of “it” in “it cancer…” Line 48: “to have” instead of “with an”, otherwise the sentence lacks of sense Line 57: “preventive” instead of “preventative” Line 237: missing “reported” or a similar word after “Previously, …” Line 262: “assay” is missing after PCR array Line 274: “Despite” instead of “Dispite” PCR array is found several times in the manuscript, but also PCR-array. Please unify the term throughout the manuscript

Author Response

We fully agree with the reviewer that the background information for metoprolol and 4-OHC should be presented in the Introduction. We have added this information to explain why metoprolol and 4-OHC were included in the assays for comparison to the Introduction accordingly (line 54-58).

Sentence 15-18, and the whole abstract, was revised to make it flow better (line 13-19 and the Abstract).

Sentence 23 in the Abstract was also rephrased according to reviewer’s suggestion.

Sentences 223-225 (line 227-229 in the new version) were also changed to improve the language.

Sentences 247-248 (line 253-255 in the new version) were changed and two references were added according to reviewer’s comments.

Sentences 258-259 (line 268-271 in the new version) were changed to improve the language.

The ROS assay in Fig.1D and Fig. 2D was separated from the other panels of the figures in purpose. The PGE2 assay (Fig.3B) was also separated from Fig.3A. In fact, the data arrangement in Fig 1-3 was not in 2x2 format because different experimental methodology was used these figures. Therefore, I think it is better to keep the format same as before.

Additional discussion and references have been added to improve the presentation (see tracked changes).

Changes have been made according to the reviewer’s “minor comments”. I appreciate very much that the reviewer has paid attention to details during the manuscript review and all the comments are very reasonable and have significantly improved our manuscript.

Reviewer 2 Report

In this article, Chen et al. examined whether carvedilol can protect murine epidermal cells and reconstituted 3D human skin against UV or H2O2-induced oxidative damage. Previous reports by these and other authors clearly demonstrated that Carvedilol attenuates solar UV radiation induced skin carcinogenesis (PMID: 28912118, 25367979). In this study, Carvedilol was shown to be effective in modulating UV-induced photodamage-associated biomarkers and gene expression profiles associated with DNA damage and repair response signaling. These results indicated that the skin cancer preventative effects of carvedilol may be partially mediated by its ROS-suppressing activity. Authors used appropriate methods. The manuscript is well written. Although, the studies are interesting, they are predominantly conducted in vitro using a single cell line and a 3D skin model. Importantly these studies appear to be extension to previous reports. It is not clear if the in-vitro doses will translate to human equivalent doses. Some in-vivo studies and use of multiple cell lines will further strengthen the manuscript. Statistics are missing for Figure 4A (Ki67 and p53 positive cells).

Author Response

We fully agree with the reviewer that the present study is in vitro using only two models and it is an extended investigation which was built upon previous studies. The objective of this study was to examine whether the antioxidant activity is one of the anticancer mechanisms of carvedilol. When we used the 3D human skin culture to verify our findings, we unexpectedly found that carvedilol can modulate DNA repair, which indicates a new mechanism. Therefore, we believe this manuscript should be a valuable addition for researchers in the field of cancer chemoprevention. I agree that further studies are needed to confirm our findings in other models in vitro and in vivo.

Although the methodology is in vitro, we expect that equivalent doses that were used in this manuscript can be tested in future clinical studies, because a topical cream formulation of carvedilol would be convenient for human use to prevent sun light induced skin cancer. Our group is currently working on this formulation according to results obtained from this manuscript, and examine its efficacy and safety, with a hope to translate the findings to humans soon.

The quantification data and statistical analysis have been added to Figure 4A. The text (line 145-151) and legend (line 175-177) has been updated.

Reviewer 3 Report

In the original manuscript entitled “The β-blocker carvedilol prevented ultraviolet mediated damage of murine epidermal cells and 3D human reconstructed skin”, the authors studied the mechanism for β-blocker carvedilol prevented UV-induced skin damage. They performed the studies in cell and mouse models, the evidences are strong. Overall their findings are novel, methodologically sound and clearly written. There are some minor points need to be addressed: 1. The rationale to use 60 mJ/cm2 need to be provided. 2. The authors should provide quantifications for Figure 4A.

Author Response

The dose of UV at 60 mJ/cm2 was chosen for the 3D model according to previous studies using the same model. This statement has been added to the Discussion (line 259-260).

The quantification data have been added to Figure 4A. The text (line 145-151) and legend (line 175-177) has been updated.

Round 2

Reviewer 2 Report

None